# Healthcare worker perspectives on adaptations to differentiated anti-retroviral therapy delivery during COVID-19 in South Africa: A qualitative inquiry

Kwena Tlhaku[1], Lindani Msimango[2], Yukteshwar Sookrajh[3], Cecilia Milford[1,4], Pedzisai Munatsi[1], Andy Gray[5], Munthra Maraj[3], Nigel Garrett[1,6], Jienchi Dorward[1,7] *

**1** Centre for the AIDS Programme of Research in South Africa (CAPRISA), University of KwaZulu-Natal, Durban, South Africa, **2** Human Sciences Research Council (HSRC), Pietermaritzburg, South Africa, **3** eThekwini Municipality Health Unit, Durban, South Africa, **4** Wits MRU, Department of Obstetrics and Gynaecology, School of Clinical Medicine, Faculty of Health Sciences, University of the Witwatersrand, Johannesburg, South Africa, **5** Division of Pharmacology, Discipline of Pharmaceutical Sciences, University of KwaZulu-Natal, Durban, South Africa, **6** Discipline of Public Health Medicine, School of Nursing and Public Health, University of KwaZulu-Natal, Durban, South Africa, **7** Nuffield Department of Primary Care Health Sciences, University of Oxford, Oxford, United Kingdom

* Jienchi.Dorward@phc.ox.ac.uk

**Data Availability Statement:** The data cannot be shared publicly due to the need to protect individual and personally identifiable information.

## Abstract

During the COVID-19 pandemic, the South African Centralized Chronic Medicines Dispensing and Distribution (CCMDD) programme, adapted to include extended 12-month antiretroviral therapy (ART) prescriptions, 3-months ART refills and earlier eligibility criteria at 6-months after ART initiation. We aimed to explore the experiences of healthcare workers (HCWs) in implementing these adaptations, and to understand the overall impact of COVID-19 on CCMDD. We conducted semi-structured in-depth interviews with HCWs in eThekwini District clinics, KwaZulu-Natal, South Africa. Interviews were audio-recorded, transcribed, translated, and analysed thematically. Between 18 February and 13 December 2022, we conducted 21 interviews with nurses, doctors, pharmacists, clinic managers and a community pick-up-point staff member. There were mixed perceptions about COVID-19 adaptations to CCMDD. HCWs reported that COVID-19 adaptations to CCMDD helped keep clients away from clinics, reducing exposure to COVID-19, minimizing queues, alleviating client frustration, and easing workload, which enabled more focused attention on clients with greater needs. Clients reportedly preferred 12-month prescriptions as it gave them independence. However, HCWs were concerned about clients' ART adherence, potential to miss out on clinical input, and difficulties aligning annual viral load results, during the 12 months without clinic attendance. The extended eligibility and multi-month dispensing were acceptable to HCWs, but concerns were expressed about non-adherence and stock shortages. Challenges, including staff shortages due to sickness, increased workload, inadequate training, HCWs' distrust in clients' ability to manage their health autonomously, and staff's limited involvement in decisions about the adaptations, impacted on their implementation. While HCWs reported benefits of 12-month prescribing, extended eligibility and multi-

Researchers interested in accessing the data from this study are requested to complete the CAPRISA data request form (https://www.caprisa.org/DBFile/Files/391bc205-fb20-4c3a-a4c5-57010e433697/REQUEST%20TO%20ACCESS%20DATA%20FROM%20A%20CAPRISA%20TRIAL.docx). The written request will be assessed by the CAPRISA Scientific Review Committee within 30 business days, and once approved, the dataset will be made available. In line with standard data access principles, CAPRISA will ensure that metadata on the datasets will be made available and anonymization and other measures will be taken to protect individual and personally identifiable information in the datasets.

**Funding:** This study is supported, in whole or in part, by the Bill & Melinda Gates Foundation (INV-051067). Under the grant conditions of the Foundation, a Creative Commons Attribution 4.0 Generic License has already been assigned to the Author Accepted Manuscript version that might arise from this submission. KT is funded by DSI-NRF Centre of Excellence for HIV Prevention (UID96354). JD was also supported by grants from the Wellcome Trust PhD Programme for Primary Care Clinicians (216421/Z/19/Z), and as an Academic Clinical Lecturer (CL-2022–13–005), is funded by the UK National Institute of Health and Social Care Research (NIHR). The funders had no role in study design, data collection and analysis, decision to publish, or preparation of the manuscript.

**Competing interests:** The authors have declared that no competing interests exist.

month dispensing in CCMDD, long-term implementation would require addressing concerns about impacts on adherence, alignment of annual viral loads and timely follow up. Prioritizing HCW input in decision-making processes and enhancing provider-client interactions will be pivotal in ensuring the effectiveness of CCMDD adaptations.

## Introduction

Providing universal antiretroviral therapy (ART) to the 8.4 million people living with HIV (PLHIV) in South Africa is challenging [1, 2], with only 5.5 million receiving ART by 2022 [3, 4]. To increase ART coverage, differentiated service delivery (DSD) models have been developed, aiming to meet clients' needs while increasing healthcare service efficiency [5].

The Centralized Chronic Medicines Dispensing and Distribution (CCMDD) programme was introduced in South Africa in 2014. The programme includes DSD models, enabling public sector patients to rapidly collect chronic medication, including ART, at alternative PuPs, such as retail pharmacies, private doctors' rooms, adherence clubs, community halls and churches, in additional to public clinics [2]. Prior to 2020, PLHIV could use CCMDD provided they were virally suppressed and on treatment for over 12 months [6]. In CCMDD, clients would attend clinics for medication prescriptions every six months, and collect medication every two months from PuPs [7]. Research conducted in a KwaZulu-Natal clinic found that collecting ART in CCMDD was acceptable to clients and healthcare workers (HCWs), but strict eligibility requirements, non-flexible ART collection dates and PuP opening hours hindered effective implementation [8].

The COVID-19 pandemic provided an opportunity to improve and expedite DSD, ensuring uninterrupted ART delivery to an increased number of clients while minimizing contact with facilities [9, 10]. Evidence regarding the impacts of the COVID-19 DSD adaptations on clinical outcomes is required, as some of these changes may have longer term benefits beyond COVID-19 [11].

In South Africa, emergency COVID-19 CCMDD adaptations included i) a temporary legislative change shifting from the legal maximum of 6-month to extended 12-month prescriptions for chronic medications [12], meaning that clients would attend clinics every 12 months for an ART prescription delivered to community PuPs, ii) having more flexible eligibility criteria, such as including people who have been on ART for six instead of 12 months, and iii) giving clients three months instead of two months ART supply [10, 13] (Table 1). There have been few studies assessing the impact of these changes on clinical outcomes in South Africa, and none evaluating HCW's perspectives on the implementation of the extended 12-month ART prescriptions in the South African context. Although the implementation of the extended ART prescription was discontinued after September 2021, 12-month prescriptions within DSD programmes and reducing clinic visits to once a year could provide a more convenient

**Table 1. Changes to CCMDD in response to COVID-19.**

|  | Previous Criteria | COVID-19 Criteria |
|---|---|---|
| *Eligibility*:<br>• Minimum time on ART | 12 months | 6 months |
| *Once referred into CCMDD*:<br>• Duration of CCMDD prescription | 6 months | 12 months |
| • Maximum ART supply at pick-up point visit | 2 months | 3 months |

service for clients and potentially reduce clinic cost and workloads [14]. Previous quantitative research showed no difference in clinical outcomes among PLHIV receiving 12-month versus those receiving the standard 6-month ART prescriptions [14].

We aimed to explore the experiences of HCWs in implementing the adaptations to CCMDD and to understand the overall impact of COVID-19 on CCMDD.

## Methods

### Study design

We used qualitative methods and employed an exploratory research design, informed by the interpretivist paradigm. This approach was chosen to understand the experiences and perspectives of HCWs in implementing adaptations to the CCMDD programme and the overall impact of COVID-19 on the programme. The reporting of the research processes in this study was guided by the Consolidated Criteria for Reporting Qualitative Research (COREQ) checklist [15].

### Study setting and participants

The study was conducted in eThekwini District, KwaZulu-Natal, South Africa. We recruited healthcare workers, clinic operational managers and members of key stakeholder organisations purposively, based on their involvement in CCMDD, from nine public health clinics (eThekwini municipal clinics) and two key stakeholder organisations, such as HIV treatment non-governmental organisations (NGO) or community organisations, in the same district. A snowball sampling technique was also used wherein research participants helped identify other potential relevant and experienced participants [16]. The clinics were purposively selected based on their involvement in CCMDD, to ensure that interviewed participants had relevant experience and insights regarding the implementation of CCMDD adaptations and the impact of COVID-19 on the programme. Additionally, some clinics were included because they were the workplaces of potential participants referred by study participants using the snowball sampling technique. Participants were 18 years and older, spoke English or IsiZulu, and were willing to provide consent.

### Data collection

Trained research assistants (KT (female, MSocSci), LM (male, MSocSci)) conducted semi-structured interviews from 18 February to 13 December 2022. We approached participants in person or telephonically, explained reasons for research and scheduled in-person or virtual interviews in English and IsiZulu, in a private room within the workplace. We used a semi-structured, open-ended topic guide (S1 File) focused on the changes to CCMDD during COVID-19 and how and why these changes had, or had not, been implemented in eThekwini. Interviewers were not previously known by participants. We aimed for 20–40 interviews but stopped at 21 when theoretical saturation was reached, meaning no new themes or insights emerged [17]. None of the approached potential participants refused to participate and no participants withdrew from the study. Interviews were 30–60 minutes, were audio-recorded, and field notes written where appropriate. Recordings were transcribed and translated from IsiZulu to English where necessary. Transcripts and findings were not reviewed by participants.

### Data analysis

Data were analysed inductively (codes were generated based on the data that was gathered) using thematic analysis [18]. KT and JD (male, DPhil) performed data coding and analysis

using NVivo qualitative data analysis software (QSR International, Melbourne, Australia). We coded research transcripts using the open and axial coding processes [19]. We formed codes from the data and categorized these codes into groups of similar descriptions as they related to the research question. Thereafter, we focused on the emerging patterns, and then collated the codes into broader themes.

## Ethics

The study received ethical approval from the University of KwaZulu-Natal Biomedical Research Ethics Committee (BREC/00002958/2021). Written or verbal informed consent (audio-recorded) was obtained from all participants.

## Results

We interviewed 21 HCWs (8 nurses, 4 doctors, 5 managers, 3 pharmacists and a community PuP worker), 76% (16/21) were female. The data analysis process revealed several key themes, including perspectives on national policy-level COVID-19 adaptations, the impact of COVID-19 on CCMDD, facility-based 'ground-level' responses to COVID-19, issues with implementation of CCMDD changes, and underlying perceptions influencing staff attitudes towards the COVID-19 policy-level adaptations.

### National policy-level COVID-19 adaptations

**6- vs 12-month ART prescriptions.** HCWs reported that 12-month prescriptions proved to be advantageous to staff and clients alike as it reduced the number of clients seen in the facility.

*"[12-month prescription] was very beneficial to both the patient and clinic itself because with the number of patients. . .coming every six months is also a strain."* (Participant 1, Female)

HCWs alluded that clients would like the 12-month prescriptions to be reinstated as it gave them independence:

*". . .the con [with 6-month scripts] is you're basically following them like a child now and they don't want this, they hate it. They want to go back to the 12-month script."* (Participant 16, Female)

They also reported that 12-month prescriptions freed up time to dedicate to clients with more needs.

*"So, the 12 months script actually worked good for us because it allowed us the time to focus on the sick ones because they were not a lot of returns."* (Participant 11, Male)

However, several HCWs worried that 12-month prescriptions increased the number of defaulters, or led to problems with adherence because clients were not monitored:

*"The problem started when they went away one year. . .the viral loads were increasing. . . So, it's obvious they were collecting, and they were because we were checking package collection and it seems as if they weren't having good behaviour cycles. . ."* (Participant 16, Female).

*". . .when we issue medication to a patient to take home and make it get very long for the patient to get reviewed, we miss a lot in between that. . .we aren't 100% sure that. . . the patient takes it accordingly at home."* (Participant 4, Female)

HCWs also revealed that 12-month prescriptions made it harder to align clinic visits with annual viral load tests, and clients sometimes forgot to come back to the facility for routine check-up because they still had treatment:

*"So, you will find out that [within] these twelve months some will be due to come back to the clinic to take bloods and. . .[they'll] not come, why? They already have treatment. . ."* (Participant 10, Female).

With the 12-month prescription, HCWs were concerned that clients would not see the need to make an extra visit to the facility to report any clinical issues:

*". . .there are lot of things that might happen to them. . .. Should they be coming back to the clinic in six months they probably will be able to report it early. Now because they know they'll be going to the clinic only in 12 months they wouldn't see the need to make that extra visit to the clinic just to report that issue. . ."* (Participant 13, Male).

**Earlier referral into CCMDD at 6- instead of 12-months after ART initiation.**   Some HCWs reported that the adjusted eligibility period of 6 months would not cause any problems because people initiating ART tend to be more stable than in the past:

*"It's rare that if it [viral load] is suppressed at 6 months, it will be unsuppressed at 12 months. . .so they can go [at 6 months into CCMDD] because they are alright. Another thing is times have changed, unlike before where we were initiating people that are very sick nowadays, we initiate people like you and I, stable."* (Participant 9, Female)

However, other HCWs felt that at 6 months, clients had not yet had enough experience taking ART to safely leave clinic care.

*"Most people at six months do not understand the treatment. You may find that the virus is suppressed, but they do not have a good adherence. . .Sometimes they will say they forgot to take pills for about three nights; sometimes, they say they were visiting somewhere where they have not disclosed that they are taking pills."* (Participant 8, Female)

HCWs also felt the 6 months eligibility adaptation did not allow for adequate monitoring of the 12-month Isoniazid Preventive Treatment (IPT) (for preventing tuberculosis), which is implemented at initiation of ART:

*"IPT takes 12 months or sometimes even longer, and in those months, you should be monitoring if they are having a bad reaction to medication or if they are not developing TB signs. So, I just feel [referring into CCMDD at] six months is too short."* (Participant 8, Female)

**3 months' ART refills.**   HCWs reported that the 3 months' ART refill was better for clients because it reduced the number of facility visits:

*". . .there's 90 days bottle. . .I think it [is for] chronic stable people who don't need to take time off at work or time to come and fetch their meds every two months or every month."* (Participant 18, Female)

However, HCWs also reported that 3 months' supply of ART had to be adjusted when they experienced stock shortages:

*". . .we faced a challenge of the shortage of treatment just after [the rollout of dolutegravir] started, we couldn't carry on with the 3 months' supply because the treatment wasn't enough, then we were giving them 2 months' supply. . ."* (Participant 9, Female)

**Overall COVID-19 adaptations.**   Overall, HCWs reported that the COVID-19 adaptations to CCMDD helped prevent and control the spread of infections in facilities:

*"I guess all worked well because we all worked out for the benefit of the patient and the facility, and to prevent the spread of COVID."* (Participant 19, Male)

HCWs also shared that the adaptations helped improve service delivery and reduced conflicts between staff and clients caused by the long waiting times:

*"[COVID-19 adaptations] has also given the nurse and patients the opportunity to save their time. The nurses' work will be up to date and the patient will get their medication on time. Also, this thing of conflicts—patients squabbling with nurses saying they have been at the clinic for long, and the nurse ends up running out of time to help patients adequately, she is tired because by 10 am she has already seen 50 patients. It is not the same anymore because now, by that time, she would have maybe seen 25 patients, she is not yet tired and annoyed, she can still continue working with a smile."* (Participant 21, Female)

## Impact of COVID-19 on CCMDD

**Staff and clients stayed away due to sickness.**   HCWs reported numerous impacts of the COVID-19 pandemic on CCMDD. Patients were afraid to come to facilities and PuPs, and facilities closed because of COVID-19 cases:

*"Patients. . .were allowed to collect meds. . . even during the lockdown, but the problem started when facilities started to shut down because of the COVID positivity rate in facilities within staff and PuPs."* (Participant 19, Male)

**Staff shortages and workload.**   HCWs reported that due to the COVID-19 pandemic, stable clients were registered onto CCMDD to decongest facilities. This increased the administrative workload of HCWs, especially where there were staff shortages due to COVID-19 outbreaks:

*"What it did create is a little bit more work. . . putting more patients onto CCMDD. . . that administrative component. . .bearing in mind that during that. . . many staff members were off on quarantine and in isolation etc."* (Participant 3, Male)

**Clients went back to rural areas and could not collect medication at PuPs.** Many CCMDD clients were displaced by the COVID-19 pandemic. People who resided in cities went back to their rural homes, and restrictions on movement during lockdown made it difficult for them to collect medication from their chosen PuPs.

*"In hard lockdown it meant that the patient is stuck far and don't have the permit to move. By the time they come back the medication has been sent back because they [PuPs] don't keep the package that is not picked for more than 14 days."* (Participant 11, Male)

One HCW reported that this also resulted in the increase in clients who had not had their CCMDD prescriptions renewed within 21 days of the previous script expiring.

**Delays in parcel delivery.** HCWs reported that many clients' medication parcels were not available at their PuPs because of delays in parcel delivery by the service provider:

*". . .it was big batches of clients that were not receiving their medication so we'd probably had 5,6 or 10 in a day that would be coming from the PuP to say that medication is not there."* (Participant 17, Female)

## Facility-based 'ground-level' responses to COVID-19

**Increased push for CCMDD during COVID-19 in 2020.** HCWs revealed that there was an increased push for the use of the CCMDD programme during COVID-19 to decongest facilities and prevent SARS-CoV-2 transmission:

*". . .we also didn't want patients coming back unnecessarily and exposing themselves unnecessarily, so CCMDD was a perfect avenue to explore, to try and prevent this congestion bottleneck that we see at the clinics, so we did probably see another increase in terms of uptake into CCMDD during that time."* (Participant 3, Male)

**Increased number of PuPs.** The increased use of CCMDD during the COVID pandemic necessitated an increase in the number of external PuPs to accommodate all decanted clients:

*"I think the registrations and identification of external PuPs is ongoing. . .But it went even quicker during CCMDD because we needed more out stations so that we could decongest facilities."* (Participant 15, Female)

In order to increase PuPs in communities, an NGO supporting CCMDD employed community members and allocated re-purposed shipping containers ("Konke containers") to which the service provider could deliver medications for dispensing.

*". . .so you are just. . .an ambitious young man in the community and [name of NGO] [provide] a sub-grant [where] they will give you a container that's like a pharmacy where peoples medication will be delivered."* (Participant 2, Female)

**Sometimes ineligible people got sent into CCMDD.** The increased pressure to decongest health facilities to prevent exposure to COVID-19 sometimes resulted in the enrolment of clients who did not meet the CCMDD eligibility criteria:

*"When the process started . . . we drew up a set of guidelines that someone had to be at 6 months [and] adequately controlled before we could send them onto CCMDD. Obviously that all got fast tracked with the increasing load and COVID etc. . .that's when I think many patients who did not meet the criteria strictly were put onto CCMDD. . ."* (Participant 13, Male)

**Staff were dedicated to ensuring CCMDD worked during COVID-19.** Clients reportedly either forgot or were unable to return to facilities for prescription renewals and routine check-up, so staff put in work to ensure they followed up with them, including calling and sending outreach teams to find them:

*". . . they just forget about return date for renewal, so we needed to remind them 48 hours before their date come, even for blood now we ended up having a list of them with their dates to come then we call to remind them."* (Participant 5, Female)

*"So, those who were due for bloods, and they couldn't come to the clinic, the outreach team would help them and do their bloods and delivered medication."* (Participant 14, Female)

**The use of cellphones as a communication tool in CCMDD.** Cellphones were an important communication tool in CCMDD during the COVID-19 pandemic.

*". . .as a clinic we were using our phones to contact patients and give them messages. On the [CCMDD] service provider's side. . .they increased their personnel to handle the call centres and tried to assist as many patients as possible. So, patients received communications from all angles."* (Participant 19, Male)

**Clinics allowed more flexibility during COVID-19.** HCWs revealed that during COVID-19, healthcare facilities were more flexible with walk-in clients, including those who were not on their records:

*". . .if you don't get your medication, what can you do? [Even if] we don't have you on [our] records, but the fact that you're saying you are on medication and you know what medication you're on, we're willing to accommodate you."* (Participant 2, Female).

## Issues with implementation of changes to CCMDD

**Including healthcare workers in decisions relating to their work.** HCWs raised concerns about the COVID-19 adaptations, indicating their limited involvement in decision-making despite being frontline workers closely connected to patient care:

*"I don't think they [policy makers] involved us. . ., they hardly involve people on the ground when they take decisions. I'm not saying they're not supposed to get regulations but sometimes get the person on the ground, we interact with these people on daily basis."* (Participant 7, Female).

**Training.** HCWs reported that in-person training was a challenge due to COVID-19 restrictions. Consequently, training sessions on the Synchronised National Communication in

Health (SyNCH) system used in CCMDD and COVID-19 adaptations were done through virtual platforms or Standard Operating Procedures (SOPs), which presented challenges:

*". . .there would be errors where a patient's appointment card is claiming that a patient has been given a one-year prescription. . . but when you look on the system the [prescriber] unfortunately made it 6 months because they were not too familiar with this. . .And unfortunately, [the CCMDD service provider] didn't get enough time to train and drill us onto this because everything just happened haphazardly. . .it just got applied and then they made some SOP and that on its own posing to be a challenge."* (Participant 4, Male).

**Staff support.** Generally, HCWs reported receiving support for implementing the adaptations from the Department of Health's (DoH) supporting partners–NGOs and external service providers. However, some felt that their senior management staff provided minimal support:

*"They'd just show to write [an] SOP that was drafted by us, and they'd just approve that it's fine whereas they were at home in their pyjamas."* (Participant 2, Female)

**Cost implications of COVID-19 adaptations.** HCWs felt there were increased expenses for the DoH and facilities when clients failed to collect medications from designated points, especially during the peak of COVID-19.

*"It became costly in terms of if you had to communicate more, it's going to cost you more, you have more people on board to do all those processes. Also remember the service provider is preparing medication and sending it to a PuP if you couldn't access a PuP you came to the facility. The facility had to give you medication so the cost implication of the service provider's cost and now the facility cost. . . during the peak COVID it has happened a lot"* (Participant 19, Male)

### Underlying perceptions shaping staff's attitudes towards adaptations

Two themes underpinned the HCWs' perspectives of CCMDD and COVID-19 adaptations.

**Clients lack independence and are irresponsible.** Some HCWs believed that clients should visit facilities frequently to ensure their well-being, they perceive that clients lack the ability to effectively manage their own health.

*"Personally, I don't see the overall benefit for the patient's well-being to come to the clinic once in twelve months. . .Why have them in our market in the first place if we want them to be independent. They should just use Dr Google."* (Participant 13, Male)

Their reflections on the implementation of COVID-19 adaptations suggested a lack of trust in clients to take responsibility for their well-being.

*". . .we needed to get a little bit of buy-in from the staff because I don't think there was this comfort of just sending patients out to collect medication. Generally speaking, patients don't keep to their dates most of them and there was a concern that if a patient is going out to collect their medication and we were not actively seeing those patients that they might drop out of care and they might default and that type of thing. . ."* (Participant 3, Male)

HCWs expressed reservations regarding the reliability of clients' self-reported treatment adherence. Consequently, clients were required to provide proof of taking medication while they were away to be reinstated into the CCMDD programme:

*". . .you'll find few they'll bring proof,—others didn't have so if you didn't have proof, we cannot rely on the word of mouth. . .we have to do bloods, screen you then those that did not have proof when you do the bloods you see this one had treatment [as they are] . . . virally suppressed. . ."* (Participant 7, Female)

However, some HCWs displayed more trust in clients and recognized that it was important to involve them in scheduling clinic appointments to avoid having missed appointments.

*"Yes, I ask you because I can't make you conform to what is said we must do, I have to ask if it works for you even with bloods. . .I ask if they'd like to take bloods on Friday and try to work around dates because the blood thing says we work over a month. . .even for review I ask if it will be fine to come on certain date a patient would maybe say no because I've already taken this day off maybe let's make it that day and comes." (Participant 2, Female)*

**Patient-provider power relations.**   Although HCWs granted clients the right to decide on clinic appointment dates convenient for them, they still exercised their authority over clients to regulate medication supply, aligning it with clinic visits to ensure that clients returned as required:

*". . .and then you give them medication that will last them that time, but you don't give them full month supply medication, we'd open medication and decant some and give the amount that will last a patient until the night before they come back and that will force them to come back."* (Participant 2, Female)

Sometimes, clients would use deception, and even hid important medical information, to try and get around restrictions in CCMDD eligibility, so that they could benefit from the convenience of quick pick-ups.

*Participant*: . . .So it happens that a patient comes to collect here [PuP] and when you take a closer look you see they are pregnant and when you ask how this is happening, they. . . *answer*. . . "oh no I can't go [to the clinic and] queue for a file when it is quicker here."

**Interviewer:** So, they do not tell the nurses that they are now pregnant?

*Participant*: No, they do not tell them that they are collecting their treatment outside the clinic, so that the nurse can deactivate them [from CCMDD] because they know they are going to be deregistered. (Participant 21, Female)

## Discussion

This study assessing the implementation of the DSD adaptations made within the CCMDD programme during COVID-19 from the HCWs' perspective, is the first to explore the implementation of these adaptations in the South African context. HCWs had mixed perspectives about the adaptations made in CCMDD procedures during the COVID-19 pandemic.

Although only introduced as a time-limited emergency exception to the existing legal requirements for all repeat prescriptions [12], 12-month ART prescriptions were reportedly acceptable to both HCWs and clients. The identified patient and HCW benefits are consistent with those reported in Malawi, where longer intervals between visits to facilities were perceived as ideal [20]. In our study, 12-month prescriptions were reportedly preferred because they gave clients independence, allowing them to be responsible for their own treatment behaviour. Reported challenges with 12-month prescriptions included increased rates of defaulters and high viral loads. However, a retrospective cohort analysis conducted by our team, which included the facilities where this research was conducted, found no evidence that 12-month ART prescriptions were of detriment to clients' clinical outcomes [14]. Other challenges reported due to this extended prescription duration, included difficulty aligning annual viral load with 12-month prescriptions, clients' forgetting to return for follow-ups, and not reporting clinical issues promptly, also found in a study where HCWs associated longer intervals between visits to facilities with poor patient health-seeking behaviour [20].

Some HCWs described the shift from a 12-month to a 6-month eligibility period as feasible, as they felt most clients newly initiated on ART tended to be stable. This is consistent with other findings that newly initiating clients are generally healthier than in the past, with higher CD4 counts, and that current antiretroviral medications are better tolerated with fewer side effects [21]. However, concerns were raised about clients' understanding of and adherence to treatment just 6 months post-initiation, and the alignment with the 12-month period of IPT. The COVID-19 pandemic saw policies updated to integrate Tuberculosis Preventative Treatment (TPT) within DSD approaches [10]. A study in Uganda reported positive outcomes of this integration, with data suggesting that differentiated care models may improve IPT completion [22].

The implementation of 3 months' ART refills was seen as a positive change, reducing clinic visits, easing burden on facilities, and minimising risk of exposure to SARS-CoV-2 infection, similar to other studies [10, 20, 23]. However, supply shortages posed challenges, reflecting global stock shortages [24]. Overall, HCWs acknowledged the positive impact of COVID-19 adaptations in preventing infections, managing facilities efficiently, and reducing client-provider conflicts caused by long waiting times.

The challenges in implementing CCMDD during COVID-19 reported by HCWs in our study are similar to those experienced in healthcare programs globally [23, 25–29]. However, expansion of the CCMDD programme was also utilised as a strategy to mitigate the impact of COVID-19, with increased referrals and the establishment of more external PuPs [7, 10], with enrolment in the national CCMDD programme increasing to approximately 4,5 million clients by June 2021 with growth of 24.5% compared to June 2020 [30]. HCWs reported employing various strategies to ensure the successful operation of CCMDD, including utilizing cellphones for timely client appointment reminders and employing outreach teams. In another evaluation of CCMDD, HCWs reported that during COVID-19, clients waited for reminders before returning to clinics for their appointments [7], highlighting that ongoing follow-up may create dependencies.

We found that the implementation of the COVID-19 adaptations to CCMDD system faced challenges. First, nurse HCWs expressed dissatisfaction with not being included in decision-making processes regarding COVID-19 adaptations. Although this might not have been possible due to the urgent nature of the pandemic, their frustration suggests that their involvement as frontline workers, could have fostered a sense of ownership and commitment to the implementation of the adaptations. This is a lesson for future changes in the evolving CCMDD programme, emphasizing the importance of allowing input from frontline HCWs. Second, due to COVID-19 restrictions, virtual training sessions and SOPs were implemented rather than in-

person sessions. Some HCWs highlighted errors and difficulties faced with the SyNCH system resulting from inadequate training, and these inconvenienced both HCWs and clients. Another qualitative study on the implementation of CCMDD prior to and during the pandemic reported consistency in the lack of adequate staff training between the two periods [7]. While support from the DoH's supporting partners was acknowledged in this study, concerns were raised about the level of 'on the ground' support from senior management staff in facilities during the pandemic and in the overall implementation of CCMDD during this time. Additionally, HCWs noted increased cost implications for the DoH and facilities, particularly during the peak of COVID-19, when clients failed to collect medications from designated PuPs.

We found that some HCWs perceived a lack of independence among clients and expressed a lack of trust in clients' responsibility for their health. HCWs sometimes demonstrated a paternalistic view of clients, highlighting that in spite of policies, the public health system can remain rigid, and unsupportive of patient-centred approaches. This may explain some of the opposition from HCWs towards the COVID-19 adaptations, which aimed to make CCMDD more flexible and accommodating for clients. Studies on clients' perspectives on DSD models report high acceptability and show preference for longer intervals between visits to facilities as ideal [7, 20]. Therefore, strategies need to be developed to address these issues to ultimately enhance the effectiveness of CCMDD and client-centered care practices.

There are some limitations to our study, including the lack of clients' perspectives. However, we interviewed different types of frontline HCWs from different health facilities and found similar experiences across interviewees (including their descriptions of clients' perceived perspectives). Our study was conducted among clinics in one urban municipality, and experiences with the implementation of CCMDD during COVID-19 may differ in and cannot be generalizable to other areas, including rural areas. Furthermore, only one community PuP staff member was interviewed. However, responses from pharmacy staff in the study were in line with those of the community PuP staff member.

## Conclusions

Despite significant disruptions to the CCMDD programme affecting clients, staff, and facilities, the challenges presented by COVID-19 spurred innovative adaptations to ensure uninterrupted ART delivery to a larger client base and minimize contact with healthcare facilities. Our findings highlighted the benefits of these adaptations as reported by HCWs. However, the sustained use of 12-month prescriptions specifically requires consideration of strategies to address HCW concerns regarding treatment adherence, alignment of annual viral load monitoring, and timely follow-up. Similar considerations will also be needed in relation to other chronic medications, as the legal restrictions apply to all repeat prescriptions for Schedule 1 to 5 medicines in South Africa. Further quantitative research is needed to fully grasp the impact of these COVID-19 adaptations on retention in care and viral loads. Further work is also needed to understand the cost implications of these adaptations, explore client perspectives on 12-month prescriptions within the South African context and the dynamics of client-provider interactions to inform more effective DSD programmes and enhance service delivery.

## Supporting information

**S1 Checklist. ISSM COREQ checklist.**
(PDF)

**S2 Checklist. PLOS' questionnaire on inclusivity in global research.**
(DOCX)

**S1 File. Interview topic guide.**
(PDF)

## Acknowledgments

We would like to thank the staff and patients at eThekwini Municipality Health Unit primary care clinics.

## Author Contributions

**Conceptualization:** Lindani Msimango, Yukteshwar Sookrajh, Munthra Maraj, Nigel Garrett, Jienchi Dorward.

**Data curation:** Kwena Tlhaku, Lindani Msimango.

**Formal analysis:** Kwena Tlhaku, Cecilia Milford, Jienchi Dorward.

**Funding acquisition:** Nigel Garrett, Jienchi Dorward.

**Investigation:** Kwena Tlhaku, Lindani Msimango, Yukteshwar Sookrajh, Cecilia Milford, Pedzisai Munatsi, Andy Gray, Munthra Maraj, Jienchi Dorward.

**Methodology:** Kwena Tlhaku, Lindani Msimango, Cecilia Milford, Nigel Garrett.

**Project administration:** Kwena Tlhaku, Lindani Msimango, Pedzisai Munatsi, Jienchi Dorward.

**Resources:** Yukteshwar Sookrajh, Pedzisai Munatsi, Munthra Maraj, Nigel Garrett, Jienchi Dorward.

**Supervision:** Cecilia Milford, Pedzisai Munatsi, Andy Gray, Nigel Garrett, Jienchi Dorward.

**Writing – original draft:** Kwena Tlhaku.

**Writing – review & editing:** Kwena Tlhaku, Lindani Msimango, Yukteshwar Sookrajh, Cecilia Milford, Pedzisai Munatsi, Andy Gray, Munthra Maraj, Nigel Garrett, Jienchi Dorward.

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
