## [Decision Letter · Decision Letter 0]

21 Jun 2024

PGPH-D-24-00792

Healthcare worker perspectives on adaptations to differentiated antiretroviral therapy delivery for people living with HIV during COVID-19 in South Africa

Dear Dr. Jienchi Dorward,

Thank you for submitting your manuscript to PLOS Global Public Health. After careful consideration, we feel that it has merit but does not fully meet PLOS Global Public Health’s publication criteria as it currently stands. Therefore, we invite you to submit a revised version of the manuscript that addresses the points raised during the review process.

We look forward to receiving your revised manuscript.

Kind regards,

Henry Zakumumpa, PhD

Academic Editor

Journal Requirements:

Additional Editor Comments (if provided):

We are delighted to share reports from our reviewers. While both reviewers find some merit in this manuscript there have made suggestions for improving the methods section (such on sample selection and issues around reporting of qualitative studies). There is a minor suggestion of changing one word in the title.

Please include a point-by-point response to each of the comments raised such that we can move swiftly to a decision.

Reviewers' comments:

Reviewer's Responses to Questions

**Comments to the Author**

1. Does this manuscript meet PLOS Global Public Health’s publication criteria? Is the manuscript technically sound, and do the data support the conclusions? The manuscript must describe methodologically and ethically rigorous research with conclusions that are appropriately drawn based on the data presented.

Reviewer #1: Yes

Reviewer #2: Yes

2. Has the statistical analysis been performed appropriately and rigorously?

Reviewer #1: Yes

Reviewer #2: Yes

3. Have the authors made all data underlying the findings in their manuscript fully available (please refer to the Data Availability Statement at the start of the manuscript PDF file)?

Reviewer #1: No

Reviewer #2: Yes

4. Is the manuscript presented in an intelligible fashion and written in standard English?

Reviewer #1: Yes

Reviewer #2: Yes

5. Review Comments to the Author

Reviewer #1: Title: I advise that you re-word your title as "Healthcare worker perspectives on adaptations to differentiated anti-retroviral therapy delivery Model in South Africa: A Qualitative Inquiry" The study group, such as people living with HIV during COVID-19, can be described in the methodology.

Study Design: Read more about qualitative studies designs and revise your design with a more appropriate qualitative design.

Results: Kindly summarize your themes and sub-themes in the background or introduction before using them in the body of your results.

Data sharing: Write a statement on data sharing and show the link where your data may be found. You can deposit your dataset in a public repository such as Mendeleys, Harvard Dataverse and share the link with the readers. Transparency in research is very important and must be upheld.

Reviewer #2: I believe this is an original research. The motive is important and timely. The researchers identified key themes that related to their original quest. The analysis was appropriately performed and findings presented in a logical and understandable manner. The arguments are logical. Overall the paper is chronological, the English grammar of the paper is fair and ethics aspects was well considered.

6. PLOS authors have the option to publish the peer review history of their article (what does this mean?). If published, this will include your full peer review and any attached files.

**Do you want your identity to be public for this peer review?** For information about this choice, including consent withdrawal, please see our Privacy Policy.

Reviewer #1: **Yes: **Dr. Omona Kizito (PhD)

Reviewer #2: No

---

## [Editor Report · Decision Letter 1]

19 Jul 2024

Healthcare worker perspectives on adaptations to differentiated antiretroviral therapy delivery for people living with HIV during COVID-19 in South Africa

PGPH-D-24-00792R1

Dear Dr Jienchi Dorward ,

We are pleased to inform you that your manuscript 'Healthcare worker perspectives on adaptations to differentiated antiretroviral therapy delivery for people living with HIV during COVID-19 in South Africa' has been provisionally accepted for publication in PLOS Global Public Health.

Best regards,

Henry Zakumumpa, PhD

Academic Editor